# Variation in Vessel Element Diameters and Densities Across Habitats at the Community and Species Levels in Southeast Florida

**DOI:** 10.3390/biology14040391

**Published:** 2025-04-09

**Authors:** George King Rogers

**Affiliations:** Horticulture Department, Palm Beach State College, Palm Beach Gardens Campus, 395 Mallard Pt., Jupiter, FL 33458, USA; rogersg515@gmail.com

**Keywords:** vessel elements, wood structure, Florida, *Myrsine cubana*, *Ximenia americana*, *Chrysobalanus icaco*, *Morella cerifera*

## Abstract

There is a rich but spotty history of research aimed at relating variations in wood structure to differences in growth conditions. Such work often centers around the structural details of vessel elements (VEs), which are hollow non-living conduits of water passage roughly comparable to pipes. The prominent characteristics of VEs are their diameters and densities (i.e., the number/mm^2^ across a stem). A broadly supported perception is a tendency for the association of comparatively small VE diameters and high densities with elevated hydraulic risk, and the reverse in relatively favorable drought-free conditions. This pattern, especially for VE diameters, is particularly well established comparing species across wetter to drier geographic regions. Comparisons across neighboring habitats sharing the same climate are limited, a deficiency addressed in the present study in South Florida, which is unstudied in this connection. Another weak area is the comparison of single species spanning different nearby habitats. The present study used microscopic examination of branchlet microtome cross-sections to address those two questions across habitats. The main findings are: 1. Shaded, well-drained hammock species on average had fewer VE/mm^2^ than exposed, poorly drained pinelands, but no significant mean difference in VE diameters. 2. Individual species across habitats displayed different modes of plasticity, some adjusting VE densities only, one species adjusting VE diameters only, and one adjusting both. Swamp species tended toward reduced VE diameters. Variation in VE densities consistently exceeded variations in VE diameters.

## 1. Introduction

The literature on comparative secondary xylem structure in relation to environmental conditions is abundant but spotty [1]. Such work tends to emphasize vessel element (VE) parameters relative to cavitation risk from hydraulic stress [2]. Central to ecological wood anatomy is a long-recognized tradeoff between VE diameter and vulnerability to stress-induced cavitation [3,4,5,6,7,8,9,10,11,12]. (A refinement to this relationship is that narrow VEs have mixed vulnerabilities [11]). Echoing concerted agreement by several authors [3,4,6,13], Ewers et al. [12] (p. 358) affirmed, “The anatomical trait with perhaps the most robust theoretical and experimental basis upon which it is linked to hydraulic function is vessel diameter”. Hence the present emphasis on that character.

Authors have paid less attention to VE density, finding this parameter to be highly variable and to have a rough negative relationship with VE diameter [1], although the two can vary independently [3]. A plot of VE densities vs. VE diameters from the present study is in the Appendix A. Carlquist [3] (p. 54) regarded VE density as an “extremely sensitive measure of mesomorphy and xeromorphy”. Stressing the contribution of the hydraulic redundancy of high density to cavitation resistance, Carlquist’s [14] “Index of Vulnerability” is calculated as VE diameter/VE density.

Published eco-anatomical projects, although considerable, are oriented largely toward geographically broad, cross-climate comparisons focused on precipitation and/or on temperatures (e.g., [1,3,4,15,16,17]). Less abundant are localized studies carried out under identical climate and seasonal conditions. The present project aims to help fill this thin spot in unstudied South Florida. A second research-deficient zone, likewise tackled in this paper, is the acknowledged [4,6] paucity of intraspecific eco-anatomical data. The present project is a series of cross-habitat comparisons in VE diameters and VE densities within small timeframes in small areas having nearly uniform seasonality and weather. Here, the main differences are in microtopography, in edaphic conditions, and in exposure.

A look at prior small-scale studies more or less similar to the present will help set the stage for the present research questions. Bass and Carlquist [18] in Israel and California encountered comparatively crowded and narrow VE in the driest habitats. Concerning sun vs. shade, in California, Carlquist and Hoekman [5] found shade-adapted riparian species to have VEs with comparatively larger mean diameters and lower mean densities than their sunny neighbors except woodland trees. Similarly, Brazilian gallery forest species had broader and fewer VEs than adjacent cerrado (savanna) species [19]. These findings centered on shade helped prompt the first research question (RQ1), listed below.

Another ecological contrast with few prior reports is flooding stress in relation to xylem traits. Woodcock et al. [20] found flood-prone Amazonian trees to have maximum VE diameters narrower than at drier sites. They suspected the narrowing to be protective from hydraulic stress due to impaired waterlogged roots. More recently, Fichtler and Worbes [21] similarly found variably stressed, including flood-stressed, tropical trees to tend toward relatively narrow VEs. These findings help prompt the second research question, asking whether Florida provides another instance of swamp-narrowed VEs.

Turning now to VE densities, Baas et al. [15] and Woodcock et al. [20] commented on the pronounced variability of this trait, even within narrow contexts. In the same vein, Oladi et al. [22] (p. 500), working in Iran on diffuse-porous *Fagus orientalis* Lipsky (Fagaceae), described VE density as, “highly affected by enviro-internal factors”. They determined the vessel size, however, to be “more static…constrained with a smaller range of variation”. These observations helped prompt the third research question comparing variation in VE diameters with VE densities.

**RQ1.** 
*In SE Florida, do woody species from shaded hardwood hammocks tend toward larger VE diameters and/or lower VE densities than species from exposed pinelands?*


**RQ2.** 
*In SE Florida within single species, do comparatively narrow VEs occur in swamps?*


**RQ3.** 
*Does proportional variability in VE density generally exceed that in VE diameters?*


## 2. Materials and Methods

Study Areas. The study area was coastal eastern Palm Beach, Martin, and (scarcely) southeastern St. Lucie counties, Florida. The mean annual rainfall at West Palm Beach in 2007–2019 was 1583 mm (Sept., 212 mm; Feb, 72 mm). The mean highs and lows were 32.3–24.4 °C in Aug and 23.3–13.8 °C in Jan. [23]. The ancient sea bottom substrate consists mostly of sand and fossil seashell fragments, with and without hardpans inland [24]. The coastal strip with scrub and hardwood hammocks is mostly well drained, except for irrelevant mangroves and shoreline flats. Inland pinelands, marshes, savannas, and swamps tend toward variably impeded drainage [24,25]. The collection localities are listed in Appendix A.

Habitats (Figure 1) were selected for being comparatively well-defined and accessible locally. Habitat selection required shrub species suitable (for microtome sectioning) spanning the habitats. Hammock (Figure 1A) was selected as shaded and relatively hydraulically favorable, as described below. Pineland (Figure 1B) is the most widespread habitat locally. Bald Cypress Swamp (Figure 1C) was included specifically to look into the reports discussed above of comparatively small VEs in swamp habitat. Scrub (Figure 1D) is a prominent, well-studied, distinctive local habitat.

Coastal scrub locally occupies nutrient-poor sand having varied depths and varied topographies, mostly rolling dunes. Scrub sites are exposed, mostly sunny, windy, and potentially excessively drained with varied sand depths. Scrub is of mixed composition, with Sand Pine *Pinus clausa* (Chapm. ex Engelm.) Vasey ex Sarg. being dominant in places, but also with codominant Saw Palmetto, along with several shrubby-to-arborescent woody dicots, including a variety of oaks (*Quercus* spp., Fagaceae), Scrub Hickory (*Carya floridana* Sarg., Juglandaceae), and additional woody angiosperm species and palms. With exceptions, most woody dicots there are dwarfed and xeromorphic. Poor nutrition may contribute to xeromorphy even where precipitation is adequate [26]. Fire may historically have occurred in scrub at varied intervals, although most contemporary local management avoids burning. An organic soil layer is thin to absent. The root systems of some scrub shrubs radiate widely, while others penetrate deeply into or perhaps through the sand. However, access to groundwater in scrub is poorly known. Several scrub plant species are parasitic or hemiparasitic, including *Ximenia americana* [27] in the present study.

**Figure 1 biology-14-00391-f001:**
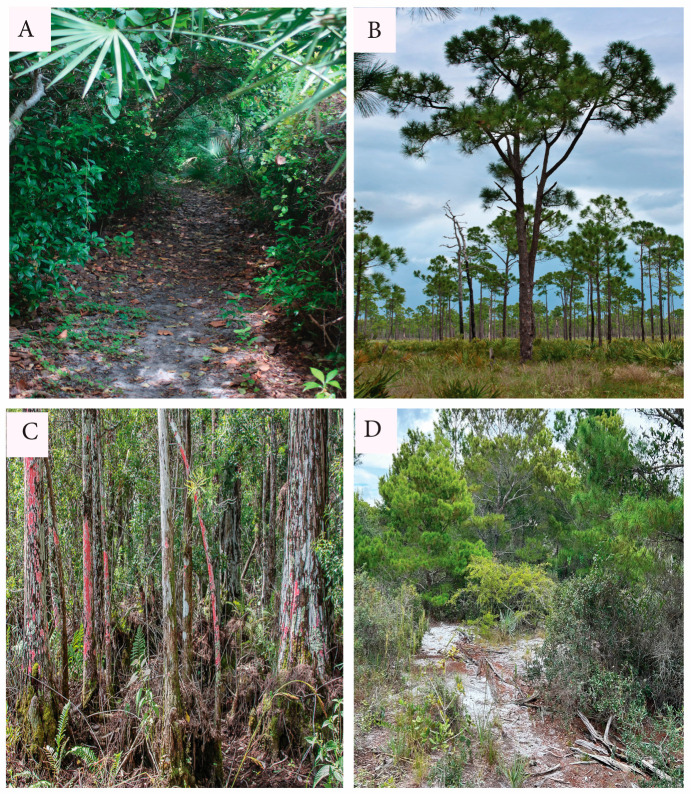
Habitats. (**A**) Coastal hardwood hammock. (**B**) Pineland (managed by prescribed burning). (**C**) *Taxodium* swamp. (**D**) Coastal scrub.

Coastal hardwood hammocks are forests under multilayered broadleaf evergreen canopies. Such hammocks are resistant to burning, tend to be well drained (with exceptions), and are protected from exposure. Fallen detritus forms an organic layer on the hammock floor. Hammocks are here regarded as particularly favorable habitats, as evidenced by their species richness. Underscoring the hydraulic favorability, this is the only local terrestrial habitat without a strong representation of conifers. Conifers tend to tolerate hydraulic stress because their wood has no VEs. Rather, the conductive cells are primarily tracheids. Tracheids confer conductive safety by tending toward small diameters, and by separation by embolism-resistant pit membranes as opposed to the open perforation plates in addition to pits on VEs [3,28,29].

Local pinelands are dominated by Slash Pine (*Pinus elliottii* Engelm., Pinaceae), with the shrubby layer containing a modest set of woody dicots and Saw Palmetto (*Serenoa repens* (W. Bartram) Small, Arecaceae). Being typically low and flat, pineland sites, including savannas and pine-dotted marshes, tend toward variably limited drainage and alternating seasonal wet and dry soil extremes [25,30]. The soil organic matter varies, often with a thick duff of pine needles, and often with a spodic soil horizon [24]. The study area pineland sites are thin-canopied to essentially fully exposed. Florida pinelands are fire-adapted and burned at different frequencies, including with prescribed burns. The shrubs there are often smaller-statured than in hammock.

The coniferous Bald Cypress (*Taxodium distichum* (L.) Rich., Cupressaceae) dominates Cypress Swamps. In the study area, only a small number of woody dicot shrubs survive on the swamp floor, especially *Chrysobalanus icaco* discussed below. These swamps are densely shaded when leafy, having filtered sunshine during the dormant season. Those locally are flooded much-to-most of the year, with the soil largely exposed late winter to late spring. That soil is carpeted with cypress needles in various states of decay. Anoxic root conditions there cause hydraulic stress [31].

Selected Species. Woody native shrubs were chosen for the single-specie surveys based on two criteria: 1. Accessibility and abundance locally in multiple divergent habitats. 2. Branchlets plentiful in the diameters used here for microtome sectioning.

*Chrysobalanus icaco* L. (Cocoplum, Chrysobalanaceae) is a robust fleshy-fruited shrub occurring in Tropical America and in Africa, as well as by introduction elsewhere. It inhabits almost every South Florida terrestrial habitat, and is a staple in commercial landscaping. *Morella cerifera* (L.) Small (Southern Wax Myrtle, Myricaceae) is a nitrogen-fixing shrub with broad natural tolerances distributed from the Southern U.S. to Central America. Its habitats range from swamps and marshes to hammocks to scrub, locally a pioneer in old fields. *Myrsine cubana* A.DC. (Colicwood, Myrsinaceae) is a large cauliflorous shrub distributed from Florida to Central America. Locally, it favors but does not require shaded moist habitats, occurring also in varied open sunny sites. *Ximenia americana* L. (Hogplum, Ximeniaceae) is a pantropical, thorny, large-fruited hemiparasitic shrub or small tree. Locally, it occurs mostly in dry open sterile habitats, abundant in scrub and in coastal hammock. Its orange drupes are roughly similar in size to the purple to white drupes of *C. icaco* (Floristic data [32,33,34]).

Methods Applied in All Surveys. Fresh branchlets were sectioned transversely usually ca. 30 µm thick on a Spencer Optical sliding microtome. Branchlet diameters were 5–10 mm diameter for the Initial Survey, mostly 7–8 mm in all the other surveys. The sections were measured as unstained water mounts at 100× magnification. VE lumen maximum diameters were determined with an ocular micrometer calibrated using a stage micrometer. (Some small-diameter conduits in cross-sections perceived as VEs may be large tracheids [3]). Study set-ups are summarized in Table 1.

**Table 1 biology-14-00391-t001:** Survey setup summary.

	Initial Survey	Repeat Survey	Myrsine Survey	Ximenia Survey	Chrysobalanus Survey	Morella Survey
Question addressed	Cross habitat difference in species mean VE diam. and/or dens.	Re-test Initial Survey	Environ. plasticity in 1 sp.	Environ. plasticity in 1 sp.	Environ. plasticity in 1 sp.	Environ. plasticity in 1 sp.
No. species studied	19 and 19 in two habitats	9 and 9 in 2 habitats	1 sp. from 2 habitat extremes	1 sp. from 2 habitat extremes	1 sp. from 2 habitat extremes	1 sp. from 2 habitat extremes
No. specimens/species	3	1	30 × 2 habitats	19 × 2 habitats	30 × 2 habitats	30 × 2 habitats
No. sections/twig	1	1	1	1	1	1
No. of VE diam. measured/section	30	30	30	30	30	30
No. VE dens. counted/section	1	4	4	4	4	1, using DotCount
Total diam./dens. measurements/survey	1650 diam./55 dens. × 2 habitats	36 dens. × 2 habitats	900 diam./120 dens. × 2 habitats	570 diam./76 dens. × 2 habitats	900 diam./120 dens. × 2 habitats	900 diam./30 dens. × 2 habitats
Locations	Several sites: (Appendix A)	School-Site, Juno-Dunes-E-Site (Appendix A)	Several sites (Appendix A)	Several sites (Appendix A)	Several sites (Appendix A)	Several sites (Appendix A)
RQ supported	1,3	1,3	2,3	1,3	1,2,3	1,2,3

A confounding circumstance is that VE parameters can vary along radial axes within individual branchlets, not necessarily as annual rings. To minimize this complication, for each survey the measurements were taken in short timeframes to negate seasonal and weather variations. Measurements were initiated at the outer edges of the xylem cylinders. Then, for VE diameters, the following protocol took place: Starting at each section’s outer ocular-view “top”, the microscope stage was moved laterally to sweep a mark on the viewfinder across the section. Every VE whose “topmost” margin appeared “below” and within 50 µm of the mark was measured. At the endpoint of each sweep, the mark was “lowered” ca. 250 µm to begin a reverse sweep. This continued, recording every VE until 30 were tabulated and averaged, rarely extended to a second section.

Cross-sectional VE density data came from counting all VEs visible in the microscope field of view at 100×, converting to mm^2^. For the Initial Survey, there was one density count per specimen, 12 o’clock at the outer wood margin. For the other surveys except *Morella* (explained below), four counts were taken at or as close as possible to the 12, 3, 6, and 9 o’clock outer wood edges.

Specific aspects of the individual surveys. During the Initial Survey, data were taken for essentially every accessible non-rare, non-vining, non-allergenic woody dicot species available within the studied habitats. Species habitat preferences (Appendix A) were assigned from three sources: Long and Lakela [32] was the primary reference. The first habitat descriptor listed there for each species was tallied as its preference. Clarification was obtained secondarily when necessary using two additional floras [33,34]. For the Initial Survey, 30 VE diameter measurements from three individuals of each species yielded 90 measurements per species (except for only a single specimen of *Stillingia aquatica* Chapm. Euphorbiaceae). One VE density count was recorded per section, three specimens/species. This community-level survey addressed RQ1 (hammock vs. pineland) and addressed RQ3 (relative plasticity in VE density vs. VE diameter). Data are in Appendix A.

Repeat Survey. The results of the Initial Survey prompted the Repeat Survey. In the Initial Survey, the mean cross-habitat VE density difference was only marginally significant. As a brief check on that concern, a small “Repeat Survey” compared VE densities at a single pineland site with a single hammock. Both sites were entered at their main trailheads, and one branchlet microsection measured for the first nine individuals of different shrub species encountered at each site. For every specimen, four VE density counts were recorded (Appendix A).

Ximenia Survey. The survey was carried out as was the Myrsine survey, except for using 19 individual plants from multiple scrub habitats and 19 from hammocks. Data are in Appendix A. This survey addressed RQ1 and RQ3.

Myrsine Survey. To look into the tentative prediction of small crowded VEs in swamps, addressing RQ2 and 3, *Myrsine cubana* sampling was from swamps vs. pinelands. VE diameters and VE densities were compared from 30 individuals from each habitat, 30 VE diameters (900 total) and four VE densities per individual (120 total) from each habitat (Appendix A).

Chrysobalanus Survey. The Chrysobalanus survey was carried out as was the Myrsine survey using 30 samples each for multiple swamps vs. 30 from multiple scrub habitats. This survey addressed RQ2 (swamp vs. scrub) and RQ3 on a single-species level. Data are in Appendix A.

Morella Survey. The Morella survey was carried out using 30 samples each for multiple swamps vs. 30 from multiple scrub habitats. Due to difficulty in counting high VE densities in this species, the survey differed from the others by counting density using the DotCount v.1.2 software [35] with the minimum size set at “12”, one count per section. These digitally tallied counts were not compared in graphs or statistics with other VE density data from other species. This survey addresses RQ2 and RQ3. Data are in Appendix A.

*Statistics*. Calculations and graphs used R [36], applying the flintyR, cvequality, and RVAideMemoire packages. Two main variables were analyzed: VE diameters and VE densities. Maximum VE diameters per species were considered also, but were set aside for offering no evident insights for present purposes. As is a common practice with clustered data [37,38], the data points for graphs and statistical tests are species means in the Initial Survey, for individual plants in the other surveys.

Permuted *t*-tests are exact nonparametric tests robust to non-normal heteroscedastic data and to small sample sizes [39]. Christensen and Zabriskie [40] cautioned against permutated tests combining skewed data (Appendix A) with unequal sample sizes; so, the sample sizes for the Initial Survey were equalized to 19 species each. For this, the original hammock species list (35 species) was cut to 19 repeatedly for multiple data runs using random species omissions. The other surveys all had equal sample sizes from the outset. Exchangeability, as is required for permuted *t*-tests, was confirmed for the Initial Survey data using flintyR version 0.0.2 2022 [41]. Because the research questions and the graphs presented the comparisons as one-sided questions, the associated *t*-tests were correspondingly one-tailed. For the Initial Survey density data, the permuted *t*-tests yielded results hovering close to the alpha value 0.05, varying slightly with different random species removals. So, the tests were run 10 times each with different random 16-species omissions. The resulting *p* value of 0.041 in Table 2 was the mean from these 10 repeats, 10K permutations each.

To compare in a unitless fashion relative dispersions of VE diameters and VE densities for RQ3 comparing variability in VE density with VE diameters, the coefficient of variation (standard deviation/mean) was calculated for densities and diameters for each survey, except the Repeat Survey. The R package cvequality (Version 0.1.3, [42]) was used to test for significant differences in that metric (Table 2).

**Table 2 biology-14-00391-t002:** Survey results. *p*-values for VE diameters and VE densities are from permuted *t*-tests. See text for details on coefficients of variation. “NA” denotes absence of diams. data in Repeat Survey.

Surveys	n	Mean Diams. (µm)	Mean Diams.	Mean Diams. (*p*-Values)	Mean Dens. (/mm^2^)	Mean Dens.	Mean Dens. (*p*-Values)	Coef. Var. (*p*-Values)	Figures
Initial Survey	19 spp. × 2 habitats	hammock 38.4	pineland 34	0.13	hammock 111.7	pineland 182.4	0.041 (see text)	Dens.: 0.83, Diams.: 0.33 (<0.0001)	Figure 2
Repeat Survey	9 spp. × 2 habitats	NA	NA	NA	hammock 70.6	pineland 204.8	0.0045	NA	Figure 3
Myrsine	30 plants × 2 habitats	swamp 36.4	pineland 35.5	0.24	swamp 95.7	pineland 126.7	<0.0001	Dens.: 0.28, Diams.: 0.14 (<0.0001)	Figure 4
Ximenia	19 plants × 2 habitats	hammock 51.8	scrub 50.3	0.32	hammock 70.8	scrub 83.0	0.01	Dens.: 0.22, Diams.: 0.18 (0.25)	Figure 5
Chrysobalanus	30 plants × 2 habitats	swamp 57.3	scrub 65.9	0.0035	swamp 24.5	scrub 26.2	0.16	Dens.: 0.24, Diams.: 0.21 0.12)	Figure 6
Morella	30 plants × 2 habitats	swamp 27.6	scrub 32.4	<0.0001	swamp 335.3	scrub 281.7	0.0014	Dens.: 0.23, Diams.: 0.15 (0.003)	Figure 7

**Figure 2 biology-14-00391-f002:**
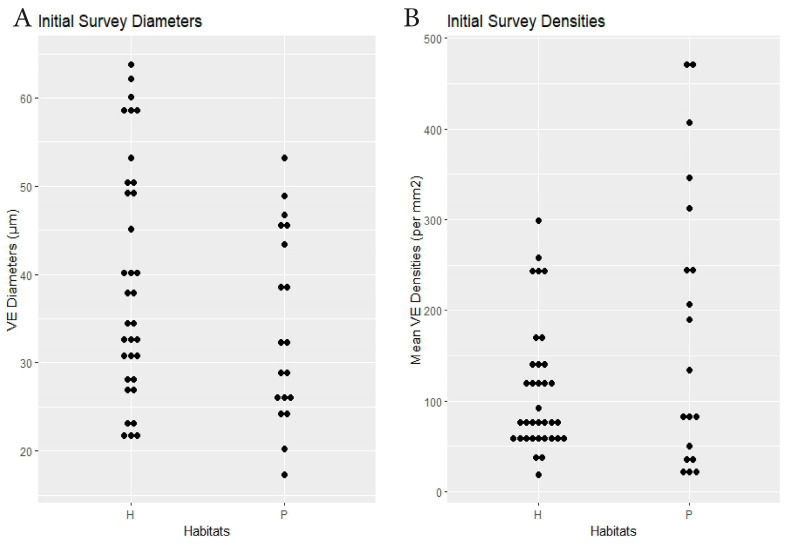
Initial survey. Habitat comparisons dotplots. (**A**) Species mean VE lumen diameters (µm). (**B**) Species mean VE densities (VE/mm^2^). H = hammock. P = pineland.

**Figure 3 biology-14-00391-f003:**
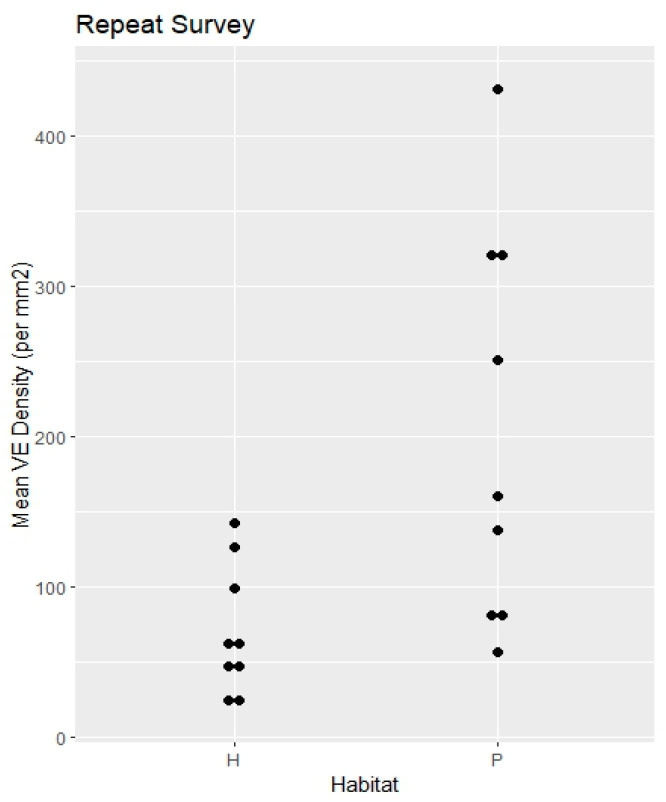
Repeat survey. Specimen mean VE densities (VE/mm^2^). H = hammock. P = pineland.

**Figure 4 biology-14-00391-f004:**
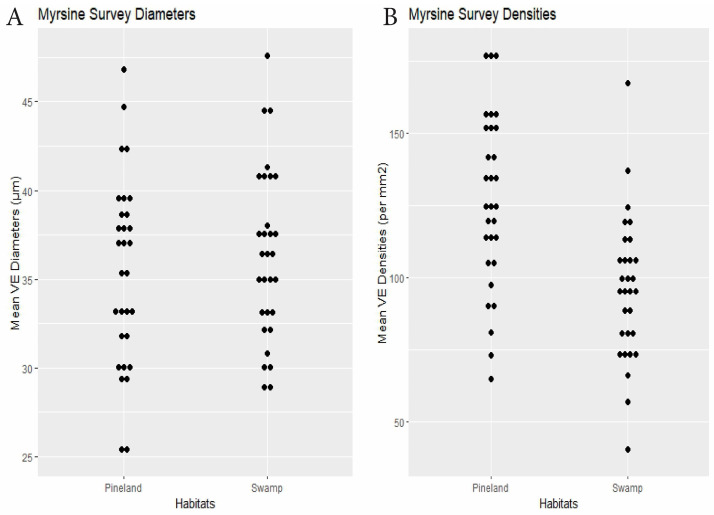
Myrsine survey. (**A**) Specimen mean VE diameters (µm). (**B**) Specimen mean VE densities (VE/mm^2^).

## 3. Results

### 3.1. Overview

VE diameters and VE densities were scattered diversely among species of different habitats. By far, the highest VE density encountered was the xeromorphic shallow-rooted shrub *Ceratiola ericoides* Michx. (Ericaceae, scrub), with a mean of 618 VE/mm^2^. The other elevated mean densities exceeding 300 VE/mm^2^ were *Lyonia lucida* (Lam.) K. Koch. (Ericaceae, National Wetland Plant List facultative wetland [43]), *Bejaria racemosa* Vent. (Ericaceae, NWPL facultative wetland), *Vaccinium myrsinites* Lam. (Ericaceae, NWPL facultative wetland), *Hypericum fasciculatum* Lam. (Clusiaceae, NWPL obligate wetland), and *Morella cerifera* (Myricaceae, NWPL facultative wetland). The ones that had the lowest mean densities, under 35 VE/mm^2^, were *Ficus aurea* Nutt. (Moraceae, NWPL facultative wetland), *Stillingia aquatica* Chapm. (Euphorbiaceae, NWPL obligate wetland), *S. sylvatica* L (FDEP facultative wetland [44]), and *Diospyros virginiana* L. (Ebenaceae, NWPL facultative wetland).

The species with the largest mean VE diameters, exceeding 60 µm, were: *Chrysobalanus icaco* (Chrysobalanaceae, NWPL facultative wetland), *Erythrina herbacea* L. (Fabaceae, FDEP upland [44]), and *Ficus aurea* (NWPL facultative wetland). Those that had mean diameters under 22 µm were: *Ceratiola ericoides* (scrub), *Hypericum tetrapetalum* (Clusiaceae, NWPL obligate wetland), *Myrcianthes fragrans* (Sw.) McVaugh (Myrtaceae, hammock), *Prunus caroliniana* (Mill.) Aiton (Rosaceae, NWPL facultative upland), and *Vaccinium myrsinites* (NWPL facultative upland). For data on individual species, see especially Appendix A.

### 3.2. With Respect to the Three Research Questions

RQ1. The hammock habitats had significantly lower VE densities in the Initial, Repeat, and Ximenia Surveys (Table 2; Figure 2, Figure 3, Figure 5 and Figure 8). No significant differences in diameters appeared in those surveys.

RQ2. Cypress swamps tended toward comparatively small VEs. In the *Chrysobalanus* survey (Figure 6 and Figure 8, Table 2), the swamp VEs had significantly narrower diameters than in scrub, without significant difference in density. In the *Morella* Survey (Figure 7 and Figure 8, Table 2), the swamp specimens had significantly smaller mean VE diameters and had larger mean densities than in the scrub. In the Myrsine Survey, the swamp specimens had no significant difference in VE diameters compared with pineland, and had lower mean VE densities (Figure 4 and Figure 8, Table 2).

RQ3. In every survey, VE density had a greater coefficient of variation than VE diameter, and the differences were significant in the Initial, Myrsine, and Morella Surveys (Table 2).

**Figure 5 biology-14-00391-f005:**
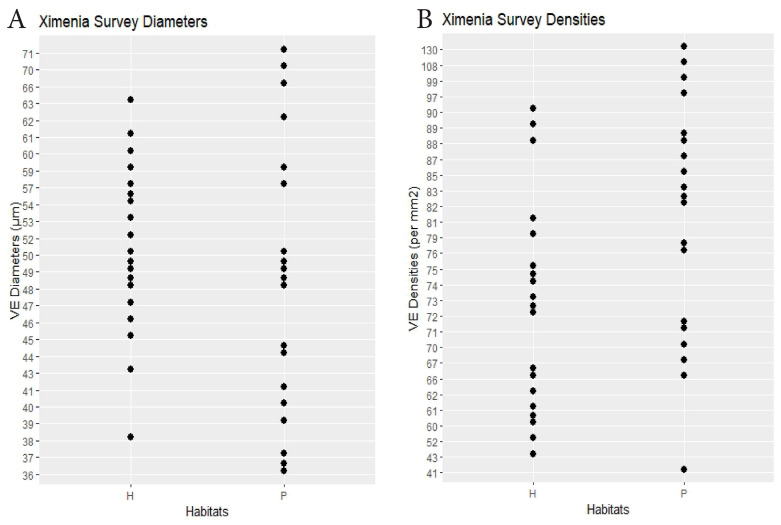
Ximenia survey. (**A**) Specimen mean VE diameters (µm). (**B**) Specimen mean VE densities (VE/mm^2^). H = hammock. P = pineland.

**Figure 6 biology-14-00391-f006:**
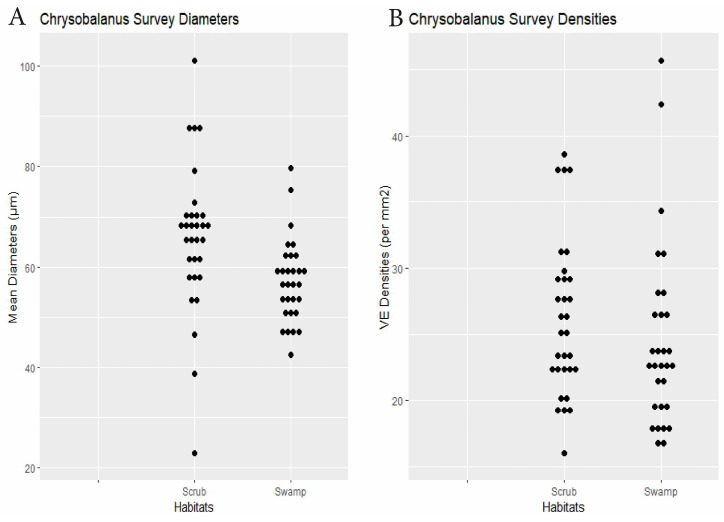
Chrysobalanus survey. (**A**) Specimen mean VE diameters (µm). (**B**) Specimen mean VE densities (VE/mm^2^).

**Figure 7 biology-14-00391-f007:**
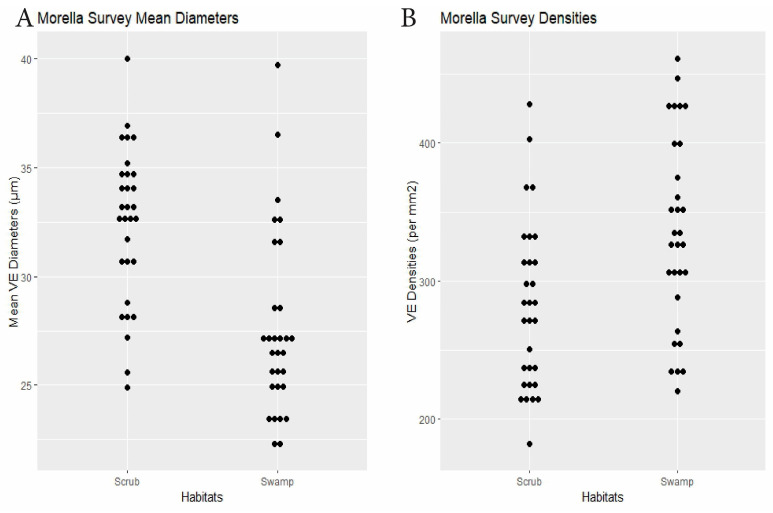
Morella survey. (**A**) Specimen mean VE diameters (µm). (**B**) Specimen mean VE densities (VE/mm^2^).

**Figure 8 biology-14-00391-f008:**
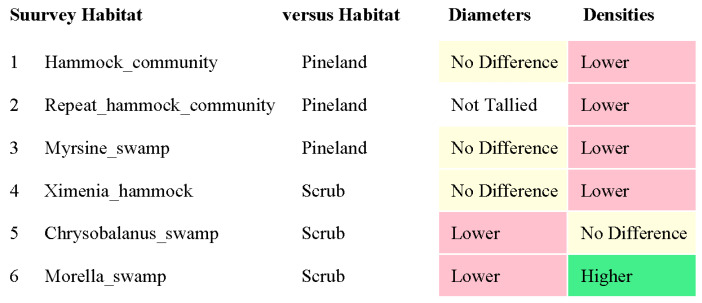
Summary of the survey results. “Lower” indicates that “Survey_habitat” (left column) had significantly lower values than “versus Habitat” for VE diameters and for VE densities. Vice-versa for “Higher”. “No Difference” means *p*-value > 0.05.

## 4. Discussion

Carlquist’s Index of Vulnerability (VE diameters/VE densities) [14] creates an expectation of relatively large VE diameters and/or relatively low VE densities in favorable habitat where conductive efficiency need not be sacrificed to safety. With respect to VE densities, this is true for the shaded hammock vs. exposed pineland habitats. This relative vulnerability in shade echoes Carlquist and Hoekman’s [5] and Sonsin et al.’s [19] similar findings for shaded habitats. These results for the present Initial community-level (and Ximenia) Surveys were curious, however, due to significant collective adjustment in VE density and not in VE diameters. This imbalance was foreshadowed by Oladi et al.’s [22] finding of VE densities being environmentally labile vs. “more-static” VE diameters. Demonstrating further the lability of VE density adjustment, Tng et al. [45] in four Australian species measured branchlet xylem adjustment to two years of artificially induced drought. In all four species, VE diameters (lumen areas) decreased as expected, but associated VE density adjustments varied substantially. In one species, VE density increased as expected with drought, but in the other three species, VE densities decreased.

Although the average adjustments in the present community-level project (and in the Ximenia and Myrsine Surveys) were mainly in VE densities, individual species in the Initial Survey showed broader cross-habitat ranges in VE diameters than others (Appendix A). Those with wide ranges included *C. icaco*, *M. cerifera*, *Hamelia patens* Jacq. (Rubiaceae), *Dodonaea viscosa* Jacq. (Sapindaceae), and *Randia aculeata* L. (Rubiaceae). The first two species in that list, surveyed individually in this paper, revealed significant adjustments in both VE densities and in VE diameters in *M. cerifera* and in VE diameters only in *C. icaco* (Table 2, Figure 8).

The significantly smaller VE diameters in swamp vs. scrub for *M. cerifera* and for *C. icaco* echoed prior studies of tropical trees in other regions [20,21]. The results of these two present surveys reveal more protective VE traits in swamp than in scrub, despite the open exposure and xeromorphic physiognomy in scrub. This hints that, within the study context, waterlogged Cypress Swamp soil may be more stressful hydraulically than upland scrub sites.

The tendency for relatively narrow VEs in swamps made the results for *C. icaco* particularly notable. While that species had narrower VEs in swamp than in scrub, it also had the broadest mean VE diameters in the study, while also arguably having the broadest habitat tolerances.

To speculate, the ability to adjust to diverse habitats and showing plasticity mainly in VE diameters as opposed to VE densities could be connected. According to Poiseuille’s law, the laminar flow rate changes with the diameter of a conduit to the 4th power but is merely proportional to the number of same-diameter conduits [6,12]. Doubling the diameter of a pipe increases its capacity 16 times in contrast to adding a second pipe to merely double the capacity. Although vessel elements are not perfect smooth pipes, adjustment in VE diameters would presumably be far more effective in adjusting hydraulic capacity than would adjustment in VE densities. However, the increased cavitational risk of broadening VE diameters, even if mitigated by mixed dimensions, may reduce the safety of upward diameter adjustment. By contrast, according to Carlquist’s Index of Vulnerability, boosting VE density adds hydraulic capacity and reduces risk. Thus, adjustment strategies featuring changes in VE diameter and/or or in VE density might offer different balances of hydraulic efficiency and safety.

## 5. Conclusions

The results reveal substantial plasticity in VE diameters and in VE densities across contrasting edaphic and biotic conditions in a small area with uniform climate. VE densities were highly variable within habitats and within species. VE density adjustment prevailed on average in the community-level comparisons, with hammocks having less-crowded VEs than pinelands. Individual species compared across habitat pairs variably showed VE density adjustment only, or VE diameter adjustment only, or a mix. Swamp habitats, probably due to root stresses from waterlogged soil, showed more protective (smaller) VE diameters than scrub habitats. VE densities consistently showed greater relative variability than VE diameters.

With an eye to future work, the present exploratory project merely scratches the surface of South Florida wood eco-anatomy. These results are a small sample of the different possible species, community, and habitat comparisons. The continued development of standardized openly accessible xylem databases such as InsideWood [46] will help broaden the comparative context. Of particular interest to extend the present work is the as-yet speculative potential relationship between modes of vascular trait adjustment to balancing conductive safety and efficiency. Beyond that, xylem parenchyma is probably relevant to the relationship of wood to environment.

## Data Availability

The original data presented in the study are openly available in https://doi.org/10.6084/m9.figshare.28661462.v1.

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
