# Peer review of "Variation in Vessel Element Diameters and Densities Across Habitats at the Community and Species Levels in Southeast Florida"

_biology, 2025, doi:10.3390/biology14040391_

Round 1
Reviewer 1 Report
Comments and Suggestions for Authors
- The study compares vessel element (VE) diameters and densities across Southeast Florida habitats, providing insights into species-specific hydraulic adjustments. The multispecies approach allows for broad ecological comparisons and species-specific responses. The study suggests that adjusting VE diameters is a more effective yet riskier strategy than modifying VE densities, but further discussion on physiological trade-offs and statistical analyses could strengthen this claim. The study contributes to understanding plant hydraulic strategies in response to habitat variation, and will contribute to a better understanding of species in contrasting habitats.
- In the simple summary the authors should consider including the gap in research and relevance of this study
- Abstract:
- A background into the study is required
- What techniques were used to compare the VE parameters?
- Please include quantitative results
- The conclusion and future studies statement should be improved.
- Introduction:
- The introduction require improvement. There is a lack of structure and flow since the authors do not give a background of the study, identifying the research gap and how their study will contribute to scientific discoveries.
- Please improve with the flow in terms of conciseness since several sentences are too long and lack references to literature
- Furthermore, there is a lack of flow since sentences are not grouped according to relevant studies
- The RQs should be grouped in a paragraph and the expected outputs should be stated.
- A simple summary on each of the selected species should be included and also justification for the selected species.
- Methods:
- Needs to be arranged logically. Table 1 cannot appear before any text description. Please include this.
- Survey results should be moved to the results.
- Results:
- can be improved with better flow and reference to tables, figures and statistics.
- Discussion
- Species names can be abbreviated once written in full for the first time.
- Remove figures from the discussion
- Please improve the discussion with a better understanding and reference to literature.
- Please include future studies in the conclusion.
Minor English editing is required.
Author Response
- In the simple summary the authors should consider including the gap in research and relevance of this study Thank you for pointing that out, this change has occurred.
2. Abstract:
1. A background into the study is required
Thank you, this change has occurred.
2. What techniques were used to compare the VE parameters?
This was added to the abstract.
3. Please include quantitative results
Principal quantitative results are now in the abstract.
4. The conclusion and future studies statement should be improved.
Thanks, this was included in the re-writing.
3. Introduction:
1. The introduction require improvement. There is a lack of structure and flow since the authors do not give a background of the study, identifying the research gap and how their study will contribute to scientific discoveries.
The introduction was fully rewritten with extensive changes, including more background, smoother flow, research gaps with the importance of the work.
2. Please improve with the flow in terms of conciseness since several sentences are too long and lack references to literature
The ms. was combed during rewriting for overly long sentences, bumpy flow, and conciseness. All of those valid corrections took place.
3. Furthermore, there is a lack of flow since sentences are not grouped according to relevant studies
Please see prior response.
4. The RQs should be grouped in a paragraph and the expected outputs should be stated.
This has taken place, thank you.
5. A simple summary on each of the selected species should be included and also justification for the selected species.
This is now in place.
4. Methods:
1. Needs to be arranged logically. Table 1 cannot appear before any text description. Please include this.
Thank you, now corrected.
2. Survey results should be moved to the results.
Done
5. Results:
1. can be improved with better flow and reference to tables, figures and statistics.
The results were rewritten with better flow, clearer reference to tables and stats, a new figure (8) added to help with this.
6. Discussion
1. Species names can be abbreviated once written in full for the first time.
Done
2. Remove figures from the discussion
Done
3. Please improve the discussion with a better understanding and reference to literature.
Yes, thanks, it has been rewritten accordingly.
1. Please include future studies in the conclusion.
Done
Thank you for these thoughtful and beneficial comments. They improved the ms.
Reviewer 2 Report
Comments and Suggestions for Authors
The MS "Variation in vessel element diameters and densities across hab-itats at the community and species levels in Southeast Florida" presents the results of extensive research aimed to reveal patterns of vessel element variation in relation to local habitat differences. Unfortunately, I found it diificult to follow the logic of the MS. On lines 61-63 as a general regularity was stated that in wet places the VE diameters are larger and densities are lower and in the driest habitats the VE are more-numerous and narrower. However, then in RQ2 it was questioned if VEs in swamps are narrower. Why? Hardwood hammocks are characterised as well drained and pinelands as poorly drained, but it was questioned "do woody species from shaded hardwood hammocks tend toward “more-mesomorphic” comparatively large VE diameters and/or low VE densities than species from exposed and poorly drained pinelands? "
In the Discussion and Conclusion this aspect was not addressed at all, although It was actually stated as one of the main question.
Minor concerns:
Line 14: benign?
Line 28 and throughout the text: - hAmmock or hUmmock?
Keywords: ecological wood anatomy and Florida wood anatomy - words are doubling; vessel element frequencies - the term is not used in the MS at all (VE density is used).
Line 277: Diameters are not more and less mesotrophic. They are large or small. The densities are higher or lower.
Author Response
Review 2
Unfortunately, I found it diificult to follow the logic of the MS.
The ms. was almost entirely rewritten with priority on flow, sequence, enhanced explanation, and overall clarity.
On lines 61-63 as a general regularity was stated that in wet places the VE diameters are larger and densities are lower and in the driest habitats the VE are more-numerous and narrower. However, then in RQ2 it was questioned if VEs in swamps are narrower. Why?
Thank you for pointing out this murkiness. The question has been addressed richly in the rewriting.
Hardwood hammocks are characterised as well drained and pinelands as poorly drained, but it was questioned "do woody species from shaded hardwood hammocks tend toward “more-mesomorphic” comparatively large VE diameters and/or low VE densities than species from exposed and poorly drained pinelands? "
Please see the prior comment, which applies here as well.
In the Discussion and Conclusion this aspect was not addressed at all, although It was actually stated as one of the main question.
Thank you for noting that, the problem is now repaired.
Minor concerns:
Line 14: benign?
Wording has been changed.
Line 28 and throughout the text: - hAmmock or hUmmock?
Hammock is correct. I think my mistake was using this Florida-specific term without adequate definition. That has been corrected.
Keywords: ecological wood anatomy and Florida wood anatomy - words are doubling; vessel element frequencies - the term is not used in the MS at all (VE density is used).
Thanks, the keyword have been replaced
Line 277: Diameters are not more and less mesotrophic. They are large or small. The densities are higher or lower.
Good point, now reworded.
Thank you for sharing these now-in-place suggestions for improvement.
Round 2
Reviewer 1 Report
Comments and Suggestions for Authors
Thank you for implementing the requested changes.
Comments on the Quality of English LanguageMinor English editing required.
Reviewer 2 Report
Comments and Suggestions for Authors
The MS has been substantially improved and became more logic.
Minor concern:
Line 26: "one" means nothong as the whole number is not mentioned. One out of 2 or 100 mean different things.